# 1-Piperidine Propionic Acid as an Allosteric Inhibitor of Protease Activated Receptor-2

**DOI:** 10.3390/ph16101486

**Published:** 2023-10-18

**Authors:** Monica Chinellato, Matteo Gasparotto, Santina Quarta, Mariagrazia Ruvoletto, Alessandra Biasiolo, Francesco Filippini, Luca Spiezia, Laura Cendron, Patrizia Pontisso

**Affiliations:** 1Department of Medicine, University of Padova, 35121 Padova, Italy; monica.chinellato@phd.unipd.it (M.C.); santina.quarta@unipd.it (S.Q.); mariagrazia.ruvoletto@unipd.it (M.R.); alessandra.biasiolo@unipd.it (A.B.);; 2Department of Biology, University of Padova, 35121 Padova, Italy; matteo.gasparotto.1@phd.unipd.it (M.G.); francesco.filippini@unipd.it (F.F.); laura.cendron@unipd.it (L.C.)

**Keywords:** G protein-coupled receptors, Protease Activated Receptor 2, allosteric modulator, 1-piperidinepropionic acid, molecular dynamics

## Abstract

In the last decades, studies on the inflammatory signaling pathways in multiple pathological contexts have revealed new targets for novel therapies. Among the family of G-protein-coupled Proteases Activated Receptors, PAR2 was identified as a driver of the inflammatory cascade in many pathologies, ranging from autoimmune disease to cancer metastasis. For this reason, many efforts have been focused on the development of potential antagonists of PAR2 activity. This work focuses on a small molecule, 1-Piperidine Propionic Acid (1-PPA), previously described to be active against inflammatory processes, but whose target is still unknown. Stabilization effects observed by cellular thermal shift assay coupled to in-silico investigations, including molecular docking and molecular dynamics simulations, suggested that 1-PPA binds PAR2 in an allosteric pocket of the receptor inactive conformation. Functional studies revealed the antagonist effects on MAPKs signaling and on platelet aggregation, processes mediated by PAR family members, including PAR2. Since the allosteric pocket binding 1-PPA is highly conserved in all the members of the PAR family, the evidence reported here suggests that 1-PPA could represent a promising new small molecule targeting PARs with antagonistic activity.

## 1. Introduction

Alterations of the extracellular environment modulate cellular activities, leading to the activation of different responses to adapt to a new condition. G-protein coupled receptors (GPCRs) play a crucial role in the transduction of such extracellular stimuli. GPCRs are endowed with broad tissue specificity and possess the ability to bind a vast range of activating molecules, including hormones, neurotransmitters, ions, photons, and odorants [1]. Despite these remarkable features, GPCRs are structurally conserved and characterized by the well-described 7-helices transmembrane domain (7TMD), connected by loops spanning the extracellular and intracellular environment [2]. The sequences and structures of these loops are crucial for GPCRs activity and specialization, as they are responsible for mediating signal transduction, even though all the five classes of GPCRs (A, B, B1, C, F) share this feature in a highly conserved fashion [3].

In recent years, structural and computational studies have significantly improved the understanding of the mechanisms of action of GPCRs and the development of new pharmaceuticals targeting them [2], especially concerning Class A, or Rhodopsin-like Class, to which belongs most of the drugged GPCRs. Due to the intrinsic allosteric nature of GPCRs, and thanks to the advancement of bioinformatic tools, it was possible to map and understand the changes that these receptors undergo to transduce the signals. Furthermore, these techniques allowed to decipher the transition from an inactive to an active state, revealing new sites for allosteric modulation [4,5]. Furthermore, they revealed that most of Class A GPCRs explore an intermediate conformation favorite by the presence of a sodium ion among the helices, coordinated particularly by TM2-3, leading the receptor to a conformation closer to the inactive state [6,7].

Among GPCRs, Protease-activated receptors (PARs) define a small subfamily, composed of four members, well-studied for their peculiar activation mechanism, which is protease-induced. As a matter of fact, these receptors are activated upon removal of their N-terminal signal peptide by protease cleavage [8,9].

The main role of these receptors has been widely described in processes related to the inflammation of the endothelium and the coagulation cascade, where they have been discovered for the first time [10]. PAR2, a member of this family, was found to play an important role in other stress responses, including cell proliferation and differentiation and its activity has been linked to epithelial-mesenchymal transition (EMT) in gastrointestinal and pancreatic cancers [11,12]. Therefore, in the last few years, many efforts have been devoted to the design of potent PAR2 inhibitors that could be applicable to cardiovascular inflammatory diseases and cancer. To reach this goal, structures of human PAR2 in complex with inhibitors have already been experimentally determined by X-ray diffraction method (PDB ID: 5NDD, 5NDZ, 5NJ6) [13,14].

In this work, the small molecule 1-piperidinepropionic acid (1-PPA) was taken into consideration for its anti-inflammatory and anticancer properties described in different patents [15,16], where it was described as an inhibitor of protein production, including cytokines and the serine-protease inhibitor SerpinB3, although its cellular target and mechanism of action are yet unclear. A recent work has reviewed the compounds targeting PAR2, highlighting the need of the development of novel PAR2 modulators to treat PAR2-mediated diseases [17]. Our goal was to investigate the capability of 1-PPA to bind PAR2 by in silico studies complemented with in vitro and cellular assays. Our results demonstrate that PAR2 represents a cellular target for 1-PPA while docked models here described let us recapitulate the modulation that such a small molecule can exert at the level of PAR2 activation and signaling.

## 2. Results and Discussion

### 2.1. CETSA Assay Suggests That 1-PPA Has a Stabilizing Effect on PAR2

To explore the hypothesis that 1-PPA can interact with PAR2 receptor, we first tested 1-piperidine propionic acid effect on PAR2 stability by the CETSA experiment, previously used to identify the AZ8838 compound as a PAR2 receptor inhibitor, with a measured antagonist potency (IC_50_) of about 2.3 µM [13,18].

CETSA is appropriate for membrane and difficult-to-purify proteins. This method relies on the possibility to determine the protein target aggregation temperature (T_agg_) by exposing cell extracts or even whole cells, overexpressing the protein of interest, to a temperature gradient. The readout obtained corresponds to the fraction of protein kept in solution at any step of the gradient. The presence of the cellular context further ensures to observe stabilization and eventual interactions in conditions resembling the physiological ones.

To set the experiment, we first determined the aggregation temperature (T_agg_) of the receptor in phosphate buffer saline (PBS) without any additional compound, which proved to be 49 °C (Figure 1a). Afterward, we performed the experiments keeping the temperature at 53 °C to ensure that more than 50% of the receptor was denatured, which let us observe the effects of the 1-PPA compound on the receptor, if any. We evaluated the effect at different compound concentrations, ranging from 2.5 to 125 µM. As shown in Figure 1b, PAR2 is more soluble at higher concentrations of 1-PPA compared to the control one in PBS only, where no protein could be detected in Dot Blot. The curve suggests that 1-PPA interacts with the protein and stabilizes its structure, maintaining it soluble at temperatures higher than that of T_agg_.

### 2.2. Molecular Dynamics Simulations Suggest That 1-PPA Occupies an Allosteric Pocket Buried within the PAR2 Receptor

To get insights into the interaction between PAR2 and 1-PPA, we applied molecular docking combined with MD simulations. The crystal structure of the human PAR2 transmembrane domain has been previously obtained in complex with the inhibitor compound AZ8838 (PDB ID: 5NDD; [1]). Since the construct used for structural studies includes extra fusion domains and few stabilizing mutations to be crystallized, we modelled *wt*PAR2 (Val61 to–Arg362) by reconstructing and refining it in silico, as described in the methods, and used the resulting model in all the subsequent computational analyses.

The putative binding poses of 1-PPA were first predicted by docking simulation. Most reliable poses were further filtered by an accessible volume analysis, performed by CASTp 3.0. Indeed, only 3 out of the 42 different solvent-accessible pockets present in the structure were shown to be accessible to 1-PPA, based on the compound volume and calculated cavities (Appendix A). Two final poses fulfilled the criteria, one at the very peripheral entrance (close to ECL2 loop) and a second one, corresponding to the pocket occupied by AZ8838 (Appendix A). As a control, this latter compound was docked to PAR2, following the unbiased docking procedure. Protocol applied here was able to predict, in the highest score cluster, the same position experimentally determined in the crystal structure (5NDD), thus supporting the robustness of the adopted procedure with an RMSD score of 0.056 between the two. Residues interacting with AZ8838 in the simulation are consistent with those identified in 5NDD, as reported in Appendix A.

Microsecond-scale MD simulations were applied both to the PAR2 receptor model and 1-PPA or AZ8838 complexes to validate docking results, and conformational rearrangements in protein structure induced by binding of those ligands were analyzed. As negative control, a third molecule (Ro5-4864) was forcefully placed in the orthosteric pocket occupied by 1-PPA and AZ8838. This compound was chosen because its size, structure and overall charge resemble those of AZ8838, but it does not belong to PAR2 known ligands [18].

Interestingly, at the end of the MD simulation, a significant shift was observed in the 1-PPA binding position, compared to the original docking pose. Indeed, 1-PPA reached an allosteric pocket, more engulfed within PAR2 transmembrane helices, roughly 9 Å apart from the initially occupied orthosteric one (Figure 2a,b and Figure 3b). Stability of structure conformation is further confirmed by PCA analysis, as reported in Appendix A. Neither AZ8838 nor the negative control underwent similar rearrangements in binding pose. While AZ8838 remained in its binding pocket throughout the whole simulation, the negative control Ro5-4864 complex proved to be unstable, hence it dissociates from the orthosteric pocket within the first 50 ns of simulation (Appendix A).

The novel pocket occupied by 1-PPA is hidden by the transmembrane helices, and the propionic moiety is placed 5 Å above the sodium cation, highly conserved among GPCRs [6,19]. Furthermore, the 1-PPA binding is stabilized both by hydrogen bonds and polar contacts in the pocket. The propionic moiety forms hydrogen bonds between its oxygen atoms and the side chains of Ser124^2.53^ and Arg121^2.50^ (The Ballesteros–Weinstein numbering scheme is based on the presence of highly conserved residues in each of the seven transmembrane (TM) helices. It consists of two numbers where the first denotes the helix, 1–7, and the second the residue position relative to the most conserved residue, defined as number 50 [20]). It is noteworthy that the Arg121^2.50^ side chain forms a hydrogen bond with one of the propionic acid oxygens, while the other coordinates Na^+^. Relevant non-bonded contacts are formed between the carbons of the piperidine ring and the side chains of residues Met159^3.36^, Phe300^6.48^, and Thr334^7.43^ (Figure 2c).

Binding of AZ8838 to PAR2 has been previously described in terms of allosteric rearrangement of ECL2 loop (residues 212–235), limiting the solvent accessibility of His227^ECL2^ [13]. Notably, a higher mobility of the ECL2 region was observed for the 1-PPA and AZ8838 bound complexes when comparing the RMSF of the holo and apo forms; however, no broad rearrangements were observed while inspecting the MD trajectories. Therefore, it cannot be excluded that the destabilization of the apo conformation of ECL2 starts in the low-microsecond scale and is completed in the mid-microsecond one (Figure 3d). In general, the observed larger mobility of the apo PAR2 compared to the holo protein states could be related to a stabilizing effect induced by the presence of the ligands, able to trap the molecule in an inactive conformation [19]. However, the Ro5-4864 bound complex does not show any mobility in the ECL2 region, which is replaced by a higher mobility of the residues 196–207 within transmembrane helix 4.

Furthermore, Figure 3c shows that both 1-PPA and AZ8838 compounds induce similar conformational adaptations in cross-correlated motions that allows describing how local protein motions in different regions of a protein are correlated. On the contrary, the pattern produced by simulation of the Ro5-4864 complex is completely different from both the apo and holo forms, further supporting the robustness of binding simulations of 1-PPA.

When considering the stability of each compound throughout the simulations, we found that while the AZ8838 compound remained tightly bound to the original pocket for all simulations, 1-PPA moved to a more buried one within the first 50 ns of simulation, where it is engaged by the previously described interactions (Figure 3b,f). Since 1-PPA detached from the initial docked pose and then remained constant for all the time of the simulation, we can deduce that such pose is the most favored one for 1-PPA binding. Of note, 1-PPA is driven by a slight reshaping of the internal cavities of PAR2 into a more continuous funnel, as an effect of slight motions of helices TM1 and TM7. Such motion is observed in all MD trajectory independently of the presence of a docked compound, suggesting it is an intrinsic property of PAR2 rather than a 1-PPA specific effect (Figure 3a).

### 2.3. 1-PPA as a Putative Pan-PAR Ligand

Given the binding and stabilizing effect of the compound described in our simulations, we decided to investigate the antagonistic properties suggested by MD using a cellular assay. To better understand the potentiality of 1-PPA as a drug, we took into consideration the high conservation of PAR family members. In particular, we focused on the degree of conservation of the residues defining the pocket occupied by 1-PPA in the four PARs isoforms (Figure 4).

The main residues forming H-bonds and establishing contacts with the ligand (Arg121^2.50^, Met159^3.36^, Phe300^6.48^, Thr334^7.43^) are all conserved or replaced by homologous amino acids except for Ser124^2.53^, which is substituted by hydrophobic residues in the other family members. Such high similarity between PARs is conserved in the hortosteric pocket occupied by AZ8838, suggesting 1-PPA might display a broad spectrum of activity toward this class of receptors and making platelets a good model to test its activity.

### 2.4. 1-PPA Antagonizes Cell Signaling and Platelet Aggregation

Firstly, HepG2 cell line was chosen as model to test the effects of 1-PPA as PAR2 antagonist. An established cell-based assay was employed to describe the phosphorylation levels of Erk1/2, since MAPKs are widely known to belong to PAR2 signaling cascade [21,22,23]. A short treatment with the specific activating peptide SLIGKV-NH_2,_ at different concentrations (1, 10, 100 µM), resulted in good levels of phosphorylated Erk1/2 (pErk1-2) already at the lowest explored concentration (Figure 5a,b). Subsequently, cells were treated with increasing concentrations of 1-PPA (0, 1, 10, 100 µM) and then activated. The amount of pErk1/2 was reduced by the presence of 1-PPA in a concentration dependent manner (Figure 5c,d).

For a more complex system, we chose the platelet aggregation assay as cellular model to unveil the properties exerted by 1-PPA on the receptor since platelets are the cell fragments with the highest level of PAR receptors on their membrane [9,24]. Cooperating with endothelial cells of the blood vessels, they start the coagulation cascade upon activation of PAR receptors by proteolytic cleavage of their N-terminal region [23]. In particular, PAR2 is activated by trypsin, tryptase, kallikreins, coagulation factors, but also by cathepsin-S or elastase.

For this study, platelets from plasma samples of healthy donors were stimulated with a thrombin substitute in presence or absence of different concentrations of 1-PPA. The presence of 1-PPA determined a progressive reduction of the level of aggregation compared to the untreated samples (Figure 5e,f). The addition of 1-PPA reduced platelet aggregation up to 50%, suggesting that this small molecule might impact the PAR receptors by blocking the cascade of events activated by such a receptor family.

## 3. Materials and Methods

### 3.1. Chemicals

1-Piperidin propionic acid (1-PPA) was purchased at SIGMA (St. Louis, MO, USA) and solubilized in Sterile PBS pH 7.4 for CETSA assays and for functional assays. SLIGKV-NH_2_ peptide was purchased from SIGMA and solubilized in DMSO for PAR2 activation.

### 3.2. Suspension Cell Culture and Transfection

The Episomal Expression System for Recombinant Protein Production in Chines Hamster Ovary Cells (ExpiCHO) was used. CHO cells were grown in suspension (160 rpm) at 37 °C and 5% CO_2_, in ProCHO5 medium (Lonza Bioscience, Basel, Switzerland) supplemented with 4 mM Glutamine (Thermofisher Scientific, Waltham, MA, USA) until they reached a concentration of 5 × 10^6^ cells/mL. Cells were then harvested by centrifugation and resuspended in fresh media at a final concentration of 2.5 × 10^6^ cells/mL. The day after, cells were transfected with 15 µg/mL Polyethylenimine hydrochloride 40,000 (FisherScientific, Hampton, NH, USA) and 3µg/mL of pcDNA3.0 plasmid, containing PAR2 FLAG and HA tagged at the N- and C-terminus, respectively (cat. #53228, AddGene, Watertown, MA, USA). Protein expression was performed by keeping the culture in constant shaking for 48 h at 30 °C in a humidified atmosphere of 5% of CO_2_ in air. Finally, cells were harvested by centrifugation (500× *g*) for 5 min at room temperature.

### 3.3. Adhesion Cell Culture

HepG2 were grown in Advanced DMEM (cat. 12491023, Thermofisher Scientific, Waltham, MA, USA) supplemented with 10% Fetal Bovine Serum (FBS) (A3840002, Gibco Thermofisher Scientific, Waltham, MA, USA) and 0.5 mg/mL of Penicillin-Streptomycin-Glutamine (10378016, Thermofisher Scientific, Waltham, MA, USA). Cells were seeded in 6 well plates and incubated at 37 °C and 5% CO_2_. The day after, cells were treated with SLIGKV-NH_2_ peptide alone or with different concentrations of 1-PPA.

### 3.4. Cellular Thermal Shift Assay

The Thermal Shift Assay (CETSA) protocol was adapted from [18]. ExpiCHO pellet was washed once with PBS pH 7.4 (137 mM NaCl, 2.7 mM KCl, 8 mM Na_2_HPO_4_, and 2 mM KH_2_PO_4_) and then centrifuged at 500× *g* for 5 min. Pellet was resuspended in PBS pH 7.4, containing protease inhibitor cocktail (539132-1SET, Sigma-Aldrich, St. Louis, MO, USA), to a final concentration of 1.5 × 10^8^ cells/mL. Samples were heated for 5 min in a thermocycler with a temperature gradient ranging from 37 to 95 °C, chilled at room temperature for 1 min, and diluted with ice-cold 1% NP-40 in PBS pH 7.4 till a final concentration of 0.2% (*v*/*v*) detergent. Samples were then exposed to 10 freeze-thaw cycles in liquid nitrogen. Finally, soluble fraction was isolated by centrifuging samples at 17,000× *g* for 30 min at 4 °C. Protein concentration was quantified by Bradford assay on the untreated control (37 °C). A total amount of 20 µg of protein, for each sample, was treated with PNGase F (P0711S, NEB, Ipswich, MA, USA) for 1 h at 50 °C to remove N-glycosylation. Samples were subjected to SDS-PAGE followed by Western Blotting.

### 3.5. Mechanistic Study

PAR2 Transfected ExpiCHO cells were resuspended as described in the previous section, equally distributed in clean PCR tubes and subjected to thermal treatment at 53 °C for 3 min. Then, 1-PPA was added to each sample at increasing concentrations (0, 2.5, 5, 12.5, 25, 50, 75, 125 µM) and samples were incubated for 2 min. Samples were then chilled at room temperature for 1 min and diluted with ice cold 1% (*v*/*v*) NP-40 in PBS pH 7.4 to a final concentration of 0.2% (*v*/*v*). Total protein concentration was quantified based on untreated control (0 µM 1-PPA) and 20 µg of total protein were treated with PNGase F. Proteins were spotted onto nitrocellulose membrane and immunoblotted as described in the next section.

### 3.6. Erk1/2 Phoshorylation

In HepG2 cells culture media was removed and replaced with sterile PBS pH7.4. PAR2 activation was performed with increasing concentrations of SLIGKV-NH_2_ activating peptide (0, 1, 10, 100 µM) for 10 min at 37 °C and 5 CO_2_. Alternatively, cells were incubated with increasing concentrations of 1-PPA (0, 1, 10, 100 µM) for 30 min and then subjected to activation with 1 µM SLIGKV-NH_2_ for 10 min at 37 °C and 5 CO_2_. Cells were washed twice with PBS to remove the exceeding compounds, scraped in fresh PBS, and collected by centrifugation for 5 min at 500× *g*. Cellular pellets were resuspended in RIPA buffer (50 mM Tris-HCl, pH 7.4, 150 mM NaCl, 0.25% deoxycholic acid, 1% NP-40, 1 mM EDTA) supplemented with protease phosphatase inhibitors cocktail (cat. 78441, Thermofisher scientific, Waltham, MA, USA) and incubated at 4 °C for 30 min.

Soluble protein fraction was obtained by centrifuging for 30 min at maximum speed at 4 °C. Protein content was estimated by Bradford assay, 20 µg of total protein was loaded for each condition on denaturing SDS-PAGE, followed by Western Blot.

### 3.7. Gel Electrophoresis and Western Blot

Protein samples were loaded on linear gradient polyacrylamide gel (SurePAGE 4-12%, GeneScript, Piscataway, NJ, USA) and were run in MES SDS Running Buffer (M00677, GeneScript) for 45 min at 180 V. Protein bands were then transferred to Nitrocellulose Membrane (Amersham Protran 0.2 NC, Cytiva, Marlborough, MA, USA) in Tris-Glycine Transfer Buffer (25 mM Tris, 192 mM Glycine pH 8.3, 15% (*v*/*v*) Methanol) applying 20 V for 1.5 h at room temperature.

### 3.8. Immunoblotting

Nitrocellulose membranes, either after SDS-PAGE transfer procedure or direct sample spotting, were blocked for 1 h in Tris Saline Buffer with Tween-20 0.05% (TTBS) and 5% (*w*/*v*) milk. Primary antibody incubations were performed at 4 °C overnight. Membranes were washed three times with TTBS and incubated with secondary antibody, HRP conjugated, for 2 h at RT (Table 1). Bands detection was performed by the addition of anti-HA (2-2.2.14, Invitrogen, Waltham, MA, USA) diluted 1:5000, while 1:5000 dilution of HL1964 (GTX637857, GeneTex, Irvine, CA, USA) was employed for Vinculin detection. pErk1/2 presence was ascertained with 9101 (Cell Signaling, Danvers, MA, USA) diluted 1:1000, while total content of Erk1/2 was detected by incubation with ERK 1/2 antibody C-9 (Sc-514302, Santa Cruz Biotechnology, Dallas, TX, USA) diluted 1:100. Bands were revealed with chemiluminescent substrate (SuperSignal™ West Pico PLUS Chemiluminescent Substrate, Thermofisher Scientific, Waltham, MA, USA) with VWR Imager CHEMI Premium (VWR). Bands intensity was quantified based on chemiluminescence count per square millimeter (I = counts per mm^2^) using ImageJ-win 64 (Fiji) software [25] and normalized to loading control. For CETSA studies, data were also normalized to the lower temperature or maximum compound concentration. Data obtained were fitted with Boltzmann sigmoidal and analyzed with “[Inhibitor] vs. response” equation in GraphPad Prism 9.5.0 (GraphPad Software, Boston, MA, USA, www.graphpad.com, accessed on 15 October 2023). Erk1/2 phosphorylation was evaluated by normalizing pErk1/2 on total levels of Erk1/2, while Vinculin was employed as reference for gel loading.

### 3.9. Protein Modelling and Docking

A model for apo PAR2 (residues 61–362) was generated by improving the crystal structure of human PAR2 in complex with AZ8838 (PDB ID: 5NDD) [13]. Firstly, experimentally introduced mutations were reverted to the wild type using PyMOL mutagenesis Wizard [26]. Then, ligands and solubilization moieties were removed and absent residues in the crystal structure were modelled with MODELLER interface in UCSF Chimera, using the DOPE-HR algorithm [27].

Docking with 1-piperidin propionic acid (1-PPA) and AZ8838 was performed by SwissDock server [28,29] with “accurate” setting for the docking and a movement freedom of 3 Å. The compounds’ topologies were automatically derived from the Merck molecular force field (MMFF) available in the CHARMM program; the protein is described by Zoete at al. [30]. Proteins were described by the CHARMM22/27 force field. Binding modes (BMs) were evaluated and clustered with FACTS [31]. For 1-PPA and AZ8838, about 250 BMs were identified for each compound. Those BMs were clustered in 39 and 41 groups, respectively. For 1-PPA, a rational selection of the most significant cluster was performed by comparing the BMs obtained with solvent accessible pockets identified with CASTp 3.0 webserver using a probe of 1.4 Å diameter [32].

### 3.10. Molecular Dynamics

Molecular dynamics (MD) was performed with Gromacs 2022.3 [33,34] using the Charmm36-jul2021 force field [35,36]. Force field parameters for AZ8838, 1-PPA, and RO54464 were derived using cgenff and manually added to complexes’ topology [37,38]. Simulations of 1 µs were performed as previously described [39]. Briefly, models were solvated with the TIP3P water model in a rectangular box with a minimum distance of 1 nm between the protein complex and the border. Simulations were performed with explicit solvent content and 0.15 M of NaCl was added to simulate physiological ionic strength. System energy was minimized by 5000 steps of steepest descent energy minimization, with a tolerance of 1000 kJ mol^−1^ nm^−1^. Subsequently, a 200 ps NVT MD simulation was used to heat the system from 0 to 310 K and equilibrated to 1 atm during a 1 ns NPT simulation. Energy minimization and equilibration simulations were performed with 1000 kJ mol^−1^ nM^−2^ on all atoms; however, they were removed for the production run. The V-rescale thermostat was used to equilibrate the temperature, whereas the C-rescale barostat was used to control the pressure [40,41]. Newton’s equation of motion was integrated using a leapfrog algorithm with a 2 fs time step. The particle mesh Ewald (PME) method was used to compute the long-range electrostatic force [42,43]. Rotational and translational motions of the system were removed, and all bonds were constrained with the LINCS algorithm. Protein conformation variations were analyzed by estimating global values of Root Mean Square Deviation (RMSD) and Root Mean Square Fluctuation (RMSF) from the original structure with Gromacs built-in rms and rmsf tools, respectively. PCA on the motion of protein–ligands complexes was performed using Gromacs built-in covar, anaeig, and sham tools. Covariance matrices were obtained from Gromacs anaeig tool and converted into cross-correlation matrices. Finally, similarity between cross-correlation matrices was assessed with Spearman rho test.

### 3.11. TRAPtest

After informed consent, 5 mL of venous blood was collected from four healthy donors after overnight fasting into syringes pre-filled with 0.5 mL of sodium citrate 109 mM with 21-gauge needles, without applying venostasis. Platelet aggregation testing was performed on the Multiplate^®^ function analyzer (Roche Diagnostics GmbH, Mannheim, Germany). The instrument continuously measures the changes of the electrical resistance (called “impedance”) between two copper wires. The greater the area, the more platelets aggregate. Briefly, 300 μL of whole citrated blood were added to an equal amount of saline solution, preheated at 37 °C, and platelet aggregation was tested after specific activation with a thrombin analogue (called TRAPtest) and with or without 1-PPA (10–1000 ng/mL) as inhibitor. Platelet aggregation was electronically measured for 6 min and expressed as units of area under the curve (AUC) plotted over time in arbitrary units (U) (1 U = 10 AU × min) [44]. Normal reference range for TRAPtest is 86–159 (U).

## 4. Conclusions

Targeting GPCR to regulate pathological conditions has been demonstrated to be one of the most valuable strategies in the pharmaceutical field in the last decades. Recent studies have shown the important role of Protease Activated Receptors, in particular of PAR2, which is involved in many inflammatory and autoimmune diseases as well as in cancer. In this work, we have demonstrated that it is possible to target this receptor with the small molecule 1-piperidinepropionic acid, previously described for its anti-inflammatory and anticancer properties.

Through direct interaction measurements, such as CETSA, we were able to describe a stabilizing effect induced by the presence of the compound, even at high temperature. Computational analysis allowed us to suggest an allosteric mechanism of action that blocks the receptor in an inactive conformation. Cell-based experiments and comparison with the other members of the PAR family allowed the disclosure of a pan-inhibitor activity across all the members, due to the high conservation of the allosteric site.

Our work opens the route to further investigations on the impact of 1-PPA activity at the cellular level. Its potential role as an allosteric inhibitor of PAR2 encourages future studies to develop potential drugs for the treatment of inflammatory conditions and cancer progression.

## Figures and Tables

**Figure 1 pharmaceuticals-16-01486-f001:**
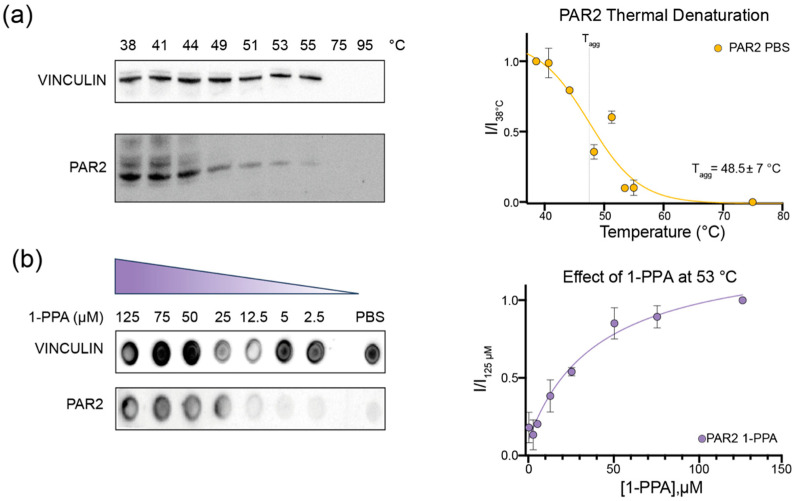
CETSA assay on PAR2 in absence and presence of 1-PPA. (**a**) PAR2 thermal denaturation performed on live cells shows a decrease in soluble protein as the temperature increases. Data of protein content were initially normalized on Vinculin content and then on the untreated sample (38 °C); (**b**) 1-PPA at increasing concentration exerts a solubilizing effect on PAR2 even at temperatures higher than the determined T_agg_. No effects are exerted on the Vinculin, which remains overall constant for each 1-PPA concentration. Therefore, data were normalized on Vinculin levels and then normalized on the highest concentration of 1-PPA (125 µM).

**Figure 2 pharmaceuticals-16-01486-f002:**
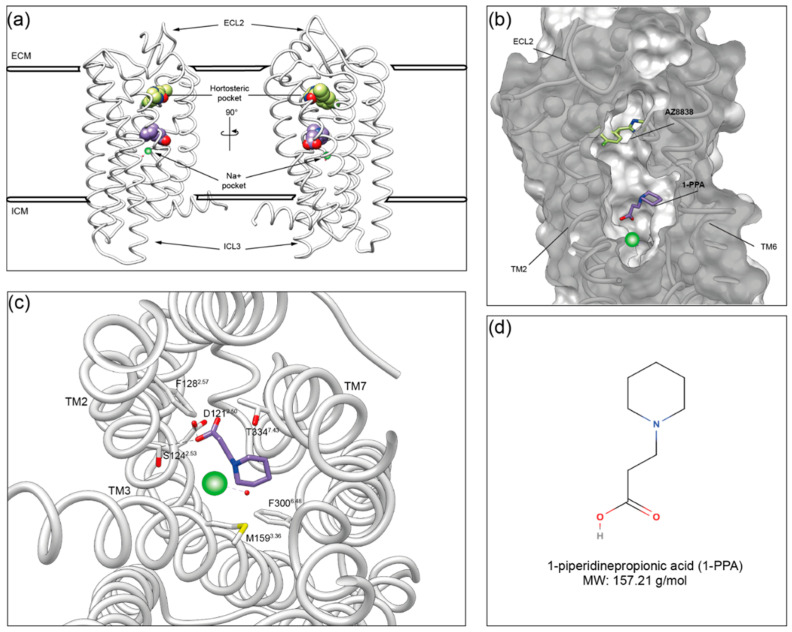
1-PPA binding site in PAR2 based on molecular dynamics. (**a**) *wt*PAR2 model of the transmembrane domain (Val61-Arg362) is in white. Among the 7TMD, 1-PPA and AZ8838 are represented in purple and light green, respectively; (**b**) Cut-through representation of the binding pockets, PAR2 surface is showing in gray while 1-PPA and AZ8838 are indicated by the arrows; (**c**) Top view of the allosteric pocket containing 1-PPA (in purple); residues interacting with it are shown as white sticks and are indicated with Ballesteros–Weinstein numeration. Hydrogen bonds formed with D121 and S124 are shown as black dashed lines. Na^+^ ion is shown in green; (**d**) Schematic representation of 1-piperidinepropionic acid structure and molecular weight (MW).

**Figure 3 pharmaceuticals-16-01486-f003:**
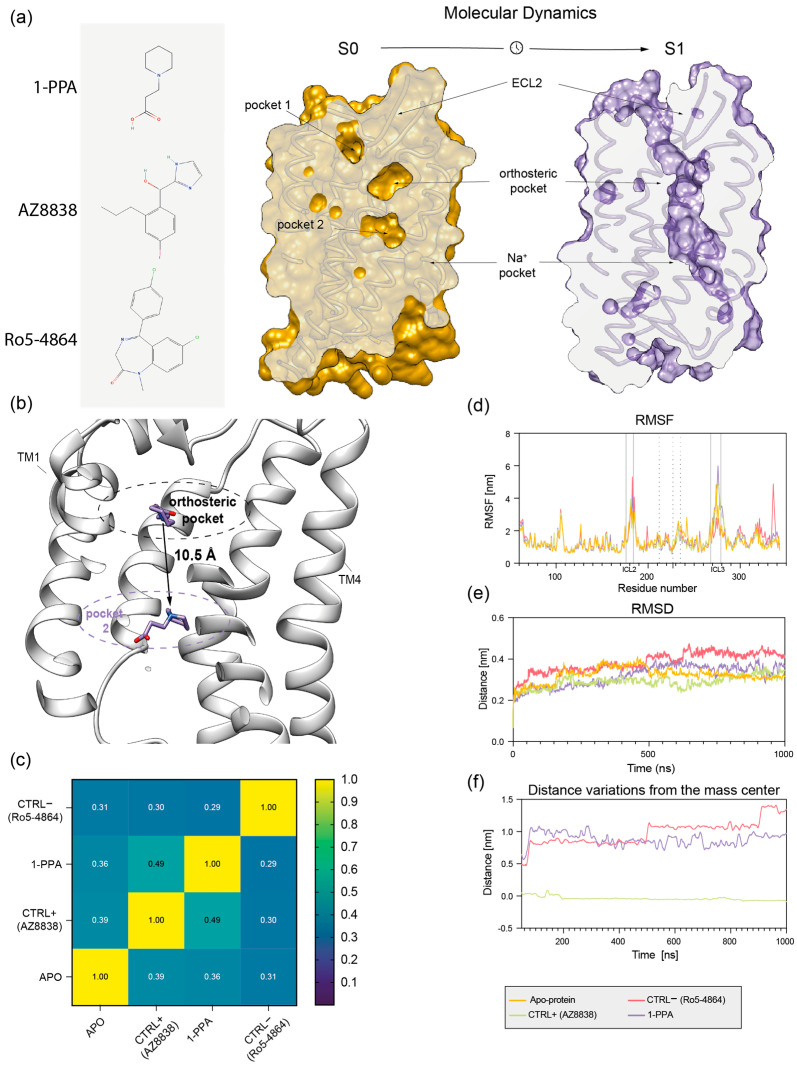
Molecular dynamics analysis. (**a**) Chemical structure of the compounds and sliced view of *wt*PAR2 model shows that three pockets, identified in the initial state (S0) in yellow, unite in a single funnel in the final state (S1), in purple; (**b**) Sliced view of wtPAR2 model, in white, shows the engulfment of 1-PPA (purple) after the simulations, in the allosteric site the by the allosteric pocket 2; (**c**) Cross-correlation matrix confronts the stabilizing effects performed by each compound during MDs; (**d**) RMSF plot representing the residue mobility, dotted lines shows the portion of the ECL2 while continuous lines box the ICL2 and 3; (**e**) RMSD variation during simulation time; (**f**) Distance variation of each compound in respect to the initial position shows that AZ8838 remains in the orthosteric pocket, while negative control Ro5-4864 and 1-PPA migrate.

**Figure 4 pharmaceuticals-16-01486-f004:**
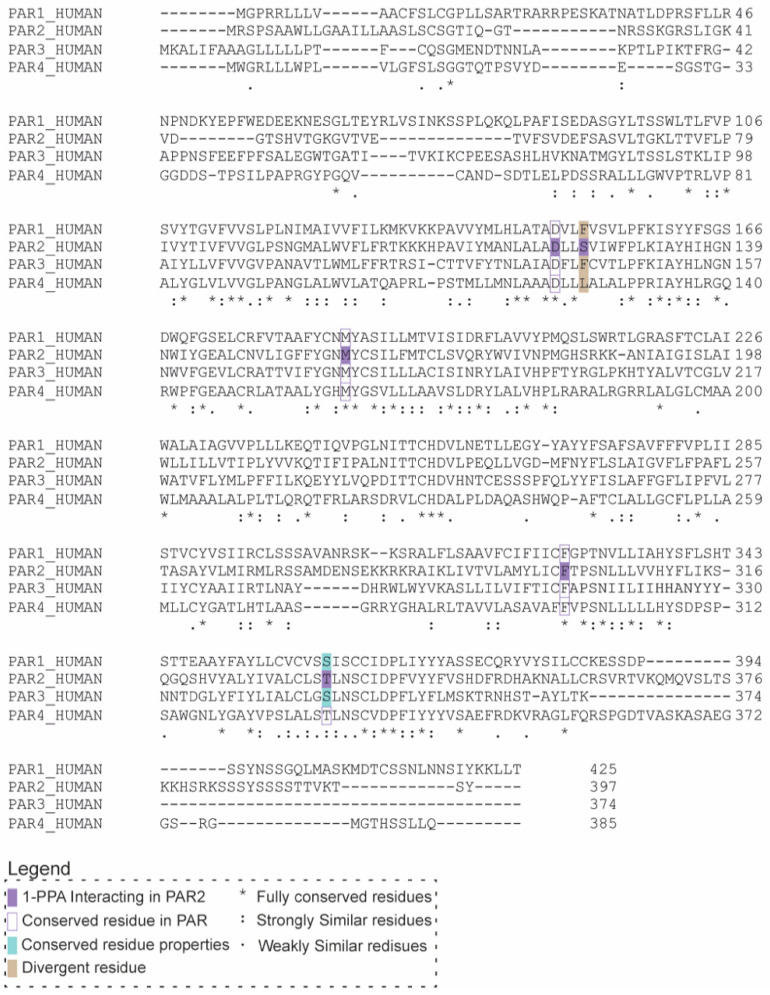
Sequence alignment of PAR family members. ClustalO alignment of sequences from Protease Activated Receptor family members. Purple boxes highlight the main residues involved in the interaction between PAR2 and 1-PPA. Lined boxes refer to residues that are fully conserved in the other proteins in the same position, while pale blue boxes identify residues conserved for their side chain properties. Gray boxes indicate residues, in the binding site, that completely diverge compared to PAR2.

**Figure 5 pharmaceuticals-16-01486-f005:**
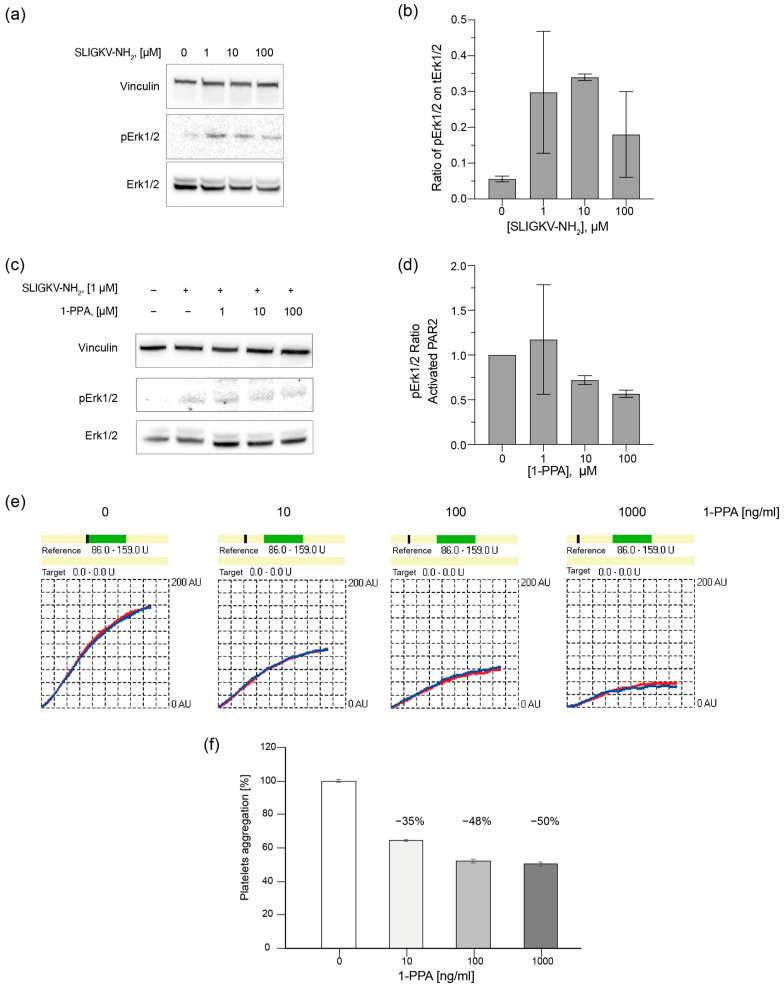
1-PPA antagonizes PARs activation. (**a**) Western Blot (WB) following the activation of PAR2 in HepG2 cell line; (**b**) Phosphorylation levels of Erk1/2 after treatment with SLIGKV-NH_2_ peptide; (**c**) WB comparing the activation of PAR2 in HepG2 cell line in presence of different concentrations of 1-PPA. (**d**) Phosphorylation levels of Erk1/2 in PAR2-activated HepG2 cells in the presence of different concentrations of 1-PPA. (**e**) TRAP test plot showing the reduction of the platelet aggregation level during time compared to the reference level of untreated platelets. (**f**) Histograms representing the percentage of platelet aggregation in presence of different concentrations of 1-PPA.

**Table 1 pharmaceuticals-16-01486-t001:** Antibodies applied for Western and Dot Blot analysis.

Target	Dilution	Code	Producer
hPAR2 (HA tag)	1:5000	2-2.2.14	Invitrogen, Waltham, MA, USA
PAR2	1:1500		
Vinculin	1:5000	HL1964	GeneTex, Irvine, CA, USA
Phospho-p44/42 MAPK (Erk1/2) (Thr202/Tyr204)	1:1000	9101	Cell Signaling Danvers, MA, USA
Total p44/42 MAPK (Erk1/2)	1:100	Sc-514302	Santa Cruz Biotechnology, Dallas, TX, USA
Mouse IgG	1:10,000	A16066	Invitrogen, Waltham, MA, USA
Rabbit IgG	1:10,000	SSA004	SinoBiological, Beijing, China

## Data Availability

Data is contained within the article and Appendix A.

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
