# Peer review of "1-Piperidine Propionic Acid as an Allosteric Inhibitor of Protease Activated Receptor-2"

_pharmaceuticals, 2023, doi:10.3390/ph16101486_

Round 1
Reviewer 1 Report
In this manuscript, the authors claim that 1-PPA acts as an allosteric ligand for PAR receptors based on cellular thermal shift assays, molecular modeling, and evaluation of platelet aggregation. While the data suggests 1-PPA may bind to PAR2, the techniques used cannot conclusively demonstrate that 1-PPA is an allosteric modulator of PAR2 or other PARs. Molecular modeling alone cannot definitively identify the binding pocket of 1-PPA - it can only predict potential binding sites. Additional experiments such as mutagenesis and functional assays are needed to validate the allosteric mechanism. As such, the conclusion that 1-PPA is an allosteric PAR inhibitor is not fully supported by the results presented.
To strengthen this study, I recommend the authors perform functional assays (e.g. Gq or β-arrestin) in the presence of a PAR2 agonist like SLIGRL-NH2 to demonstrate allosteric modulation of peptide-induced activation by 1-PPA. The modeling could also predict several putative binding sites that are then validated experimentally through mutagenesis and binding/functional assays.
While this study provides initial evidence that merits further investigation, the current manuscript draws conclusions that outpace the data. Additional mechanistic experiments are needed to conclusively validate 1-PPA as an allosteric modulator of PAR receptors. In its current form, the study provides thought-provoking but preliminary findings whose implications for PAR allosteric inhibition require more rigorous confirmation. Addressing these limitations would strengthen the conclusions and potential impact of this work.
Reasonable
Reviewer 2 Report
The manuscript presented by the authors contain interesting and important data from modeling to experimental studies including biophysical studies and also investigation of functional effects. Only two comments/suggestions should be addressed that might generate new experiments.
- the derivatization of the tool compound would be interesting, and probably meaningful to see the effect of the presence of esters instead of the carboxylic acid, the change in chain length or ring size
- usually in GPCRs there is a water network to be found in the channel that is a pocket for the tool compound. The authors might add some modeling of the change in this water network or discuss the presence of waters.
Reviewer 3 Report
The manuscript focuses on 1-PPA, described as the effective molecule in inflammatory processes and suggested to target PAR2. The authors performed the cellular thermal shift assay, molecular docking, and molecular dynamics simulation of the free ligand target and its complexes with 1-PPA and AZ8838 (control compound). The results obtained were compared with those observed in the crystallographic structures, described in terms of the binding of the small molecules and their position in the binding pocket of the target. In general, the idea of research is good, the methods chosen for such a study are appropriate. Nevertheless, the manuscript suffers from many problems which considerably diminish its quality.
I have highlighted in yellow the most critical words/phrases in the text of the manuscript and provided comments/suggestions. I hope that they will be useful for the improvement of the manuscript by the authors. The annotated version of the manuscript will be submitted as an attachment.

Extensive editing of English language required.
Round 2
Reviewer 1 Report
The authors have addressed my major concern. I recommend to publish it.
Author Response
We are grateful to the Reviewer for accepting our modified version of the manuscript.
Reviewer 2 Report
Authors answered my second question correctly and the manuscript was improved significantly. However, my first questions was absolutely not answered, the authors claim that this is our of their scope. However, in a presigeous journal like pharmaceuticals it would be suggested to show more derivatives. i let this issue to the editor's decision.
Author Response
We acknowledge the comments of the Reviewer 2, who defers to the Editor’s decision for his first question.